# Application of Self-Assembly Nanoparticles Based on DVDMS for Fenton-Like Ion Delivery and Enhanced Sonodynamic Therapy

**DOI:** 10.3390/bios12040255

**Published:** 2022-04-18

**Authors:** Jinqiang Liu, Shiying Fu, Jiaxuan Xie, Jianzhong Zhang, Jintao Pan, Chengchao Chu, Gang Liu, Shenghong Ju

**Affiliations:** 1Jiangsu Key Laboratory of Molecular and Functional Imaging, Department of Radiology, Zhongda Hospital, Medical School of Southeast University, Nanjing 210000, China; 220193657@seu.edu.cn; 2State Key Laboratory of Molecular Vaccinology and Molecular Diagnostics & Center for Molecular Imaging and Translational Medicine, School of Public Health, Xiamen University, Xiamen 361102, China; fushiying@stu.xmu.edu.cn (S.F.); xiejiaxuan@stu.xmu.edu.cn (J.X.); jzzhang@stu.xmu.edu.cn (J.Z.); jintaopan@xmu.edu.cn (J.P.)

**Keywords:** sonodynamic therapy, chemodynamic therapy, theranostics, DVDMS, self-assembly

## Abstract

Upon harnessing low-intensity ultrasound to activate sonosensitizers, sonodynamic therapy (SDT) induces cancer cell death through the reactive oxygen species (ROS) mediated pathway. Compared with photodynamic therapy (PDT), SDT possesses numerous advantages, including deeper tissue penetration, higher accuracy, fewer side effects, and better patient compliance. Sinoporphyrin sodium (DVDMS), a sonosensitizer approved by the FDA, has drawn abundant attention in clinical research, but there are some deficiencies. In order to further improve the efficiency of DVDMS, many studies have applied self-assembly nanotechnology to modify it. Furthermore, the combined applications of SDT/chemodynamic therapy (CDT) have become a research hotspot in tumor therapy. Therefore, we explored the self-assembly of nanoparticles based on DVDMS and copper to combine SDT and CDT. A cost-effective sonosensitizer was synthesized by dropping CuCl_2_ into the DVDMS solution with the assistance of PVP. The results revealed that the nanostructures could exert excellent treatment effects on tumor therapy and perform well for PET imaging, indicating the potential for cancer theranostics. In vitro and in vivo experiments showed that the nanoparticles have outstanding biocompatibility, higher ROS production efficiency, and antitumor efficacy. We believe this design can represent a simple approach to combining SDT and CDT with potential applications in clinical treatment and PET imaging.

## 1. Introduction

Sonodynamic therapy (SDT), an emerging noninvasive clinical treatment modality for cancer therapy, activates sonosensitizers using low-intensity ultrasound and produces reactive oxygen species (ROS) to induce cancer cell death [1,2]. Compared with photodynamic therapy (PDT), SDT has many advantages, including deeper tissue penetration, higher accuracy, fewer side effects, and better patient compliance. Recently, some clinical data have been reported on the use of SDT as an adjuvant therapy in breast cancer [2]. Among SDT, low-intensity ultrasound activates sonosensitizers to generate ROS, which triggers cancer cell death through apoptosis and/or necrosis. The mechanism of ROS production in SDT remains to be further studied, but it is widely assumed to rely on several known mechanisms, such as sonoluminescence, pyrolysis, ROS production by the collapse of cavitation bubbles, and ROS-dependent cytotoxicity [3]. Substantial efforts have been devoted to accomplishing sufficient ROS production, however, the development of novel sonosensitizers with efficient sonodynamic efficacy remains a considerable challenge [4].

Most of the known sonosensitizers are originally extracted from photosensitizers used in PDT, such as hematoporphyrin, protoporphyrin, and near-infrared (NIR) dyes, some of which are derived from natural plants. Nevertheless, such photosensitizers have great limitations including skin sensitivity, potential phototoxicity, poor pharmacokinetics, and distribution [5]. An ideal sonosensitizer should be safe, easy to use, have low toxicity or nontoxicity, high sensitivity, excellent tumor cells selectivity, rapid clearance in normal tissues, and no permanent damage to fundamental physiological behaviors, such as porphyrins, phthalocyanines (Pcs), antitumor drugs, nonsteroidal anti-inflammatory drugs, and xanthene derivatives [6]. What is more, some inorganic sonosensitizers have been researched [7,8,9]. Wang X. et al. synthesized the ultra-small titanium nitride (TiN) nanodots for photothermal-enhanced SDT [10]. Sinoporphyrin sodium (DVDMS) is an active component extracted from Photofrin and has been approved by the FDA [11], but its efficacy is 10 times greater than that of Photofrin. DVDMS can be used as a sonosensitizer, which shows favorable efficacy not only in tumor therapy but also in treating infectious diseases. Currently, two clinical trials of DVDMS are underway (No. CTR20150690 and CTR20150725) [12]. However, as a small molecule drug, DVDMS exists some disadvantages, such as insufficient efficiency, low concentration in new blood vessels of tumors, and high toxicity in normal tissues. In order to further optimize the efficiency of DVDMS, nanotechnology has been applied in many studies. Some studies exploit the high affinity between DVDMS and proteins to attach DVDMS to the exosome surface or coat it in the exosome core, showing ultrasound-responsive controlled release and enhanced SDT [13].

Additionally, self-assembly induced by metal ions is also a research hotspot and allows for the facile combination of imaging and therapy. Multi-functional nanocomposites can be obtained by inducing self-assembly among multiple components and further modification to realize effective integration of diagnosis and treatment [14,15]. Studies found that utilizing metal to realize DVDMS self-assembly was an impressive attempt to improve the SDT efficiency. Some researchers chelated manganese ions with DVDMS to form nanoparticles to enhance the SDT effect [16]. In addition, it was found that metal oxides could also be assembled with DVDMS to enhance treatment efficacy [17]. Studies found that SDT mediated by DVDMS produced its effects mainly by inhibiting the PI3K/AKT/mTOR signaling pathway [18].

Chemodynamic therapy (CDT) refers to a treatment that utilizes Fenton/Fenton-like reactions to generate ·OH in the tumor [19]. Some studies found that some metal ions, including Fe^2+^, Mn^2+^, Co^2+^, Zn^2+^ and Cu^2+^, exhibited Fenton/Fenton-like activity [20,21]. The combined application of SDT/CDT is one of the most promising methods for future tumor therapeutics [22]. Sun Y. et al. synthesized the AIPH@Cu-MOF, combining SDT and chemodynamic-induced free radical generation in an oxygen-independent manner [23]. To address the insufficient efficiency and promote the endocytosis capacity of DVDMS, we explored the self-assembly nanoparticles based on DVDMS and copper ions to combine SDT and CDT. Cu/DVDMS demonstrated superior SDT properties relative to free DVDMS. When Cu/DVDMS entered the cytoplasm of cancer cells, Cu^2+^ was converted into Cu^+^ by GSH, due to the higher level of glutathione in tumor cells. Then, Cu^+^ could continue to react with intracellular H_2_O_2_ to generate highly toxic ·OH and cause DNA damage, protein inactivation, lipid oxidation, and apoptosis [24]. In addition, ^64^Cu is a new type of PET imaging nuclide. PET has been widely utilized in clinical settings owing to its high sensitivity, good spatial resolution, and accurate quantification [25]. Taken together, the designed Cu/DVDMS nanoparticles have potential applications in clinical treatment and PET imaging.

## 2. Materials and Methods

### 2.1. Preparation of Cu/DVDMS Nanoparticles

Simply, 10 mL CuCl_2_ (Aladdin, Shanghai, China) solution and PVP (Aladdin, Shanghai, China) (5% The mass fraction) were added into 10 mL DVDMS (Jiangxi Qinglong Group Co., Yichun, China) solution (100 μg/mL) ([Cu^2+^]/[DVDMS] ratio = 5:1) and stirred for 3 h at room temperature. The nanoparticles were obtained after centrifugation-filtration through 30 kDa MWCO Amicon filters and gradient centrifugation. The structure and morphology of the nanoparticles (NPs) were captured by transmission electron microscope (TEM, Olympus, Tokyo, Japan). The surface zeta potential and dynamic light scattering (DLS, Malvern, Worcestershire, UK) size of the NPs were detected. UV-vis-NIR absorbance spectra of Cu/DVDMS were recorded using a microplate reader (Thermo scientific, Waltham, MA, USA). The X-ray photoelectron spectroscopy (XPS) analysis of the NPs was detected by an X-ray photoelectron spectrometer (Oxford instruments, Oxford, UK).

### 2.2. SDT Property of Cu/DVDMS

2′,7′-Dichlorodihydrofluorescein (DCFH, Beyotime biotechnology, Shanghai, China) probe was applied to detect the ROS production from the Cu/DVDMS where the DVDMS was equivalent to 100 μg/mL. Firstly, 10 μL of DCFH-DA (10 mM) was added to 30 μL NaOH solution (10 mM), and the reaction mixture was kept in dark for 30 min. Afterward, 2 mL PBS (pH 7.4) was added to neutralize the reaction. Secondly, 100 μL DCFH was added to the Cu/DVDMS mixture. Thirdly, the mixture was exposed to ultrasound (US) (1 MHz, 1 W/cm^2^, 50% duty cycle, 3 min) for 3 min. By measuring the fluorescence intensity of the DCFH-DA at Ex 488 nm, ROS production could be estimated.

### 2.3. CDT Property of Cu/DVDMS

To study the CDT property of Cu/DVDMS, methylene blue (MB, Sinopharm, Beijing, China) was applied to detect the hydroxyl radical (•OH) of Fenton-like reaction with the Cu/DVDMS where the DVDMS was equivalent to 100 ug/mL. Cu/DVDMS solution or DVDMS solution (100 μg/mL), GSH (Sigma, St. Louis, MO, USA) solution (10 mM), H_2_O_2_ solution (10 mM), and MB solution (33.3 μM) were mixed and stirred in dark for 1 h. CDT property could be estimated by detecting the UV-vis-NIR absorbance spectra.

### 2.4. Cellular Uptake of Cu/DVDMS

The 4T1 cancer cells were seeded into the confocal dish at a density of 10^4^ cells/well and cultured for 12 h. DVDMS and Cu/DVDMS were added and co-cultured separately for 3, 6, and 12 h. Then, the free DVDMS or Cu/DVDMS were wiped out by removing the culture medium and washed with PBS for three times. Paraformaldehyde (POM, Sigma-Aldrich, St. Louis, MO, USA) was used to fixate the cells for 15 min and removed. For the fluorescence imaging study, a solution containing 4′,6-diamidino-2-phenylindole (DAPI, Sigma-Aldrich, St. Louis, MO, USA) was added to the cells and the cells were observed by a confocal laser scanning microscope (CLSM, Olympus, Tokyo, Japan). Copper ions quantitatively were measured by inductively coupled plasma atomic mass spectrometry (ICP-MS, SPECTRO, Massachusetts, USA) at 1, 3, 6, and 12 h. The 4T1 cancer cells incubated with DVDMS or Cu/DVDMS for 12 h were collected for TEM.

### 2.5. Intracellular ROS Production of Cu/DVDMS 

The intracellular ROS levels of different treatment groups were detected using DCFH-DA probe. Firstly, the 4T1 cancer cells were seeded into several 24-well plates with the density of 2.5 × 10^5^ cells/well and cultured for 12 h. The cells were treated with DVDMS or Cu/DVDMS for 12 h. Then, the cells mixed with DCFH-DA (50 μM, 15 min) were exposed to US (1 MHz, 1 W/cm^2^, 50% duty, 3 min). Lastly, the cells were observed by fluorescence microscope (Olympus, Tokyo, Japan) and analyzed by flow cytometry analysis.

### 2.6. Biocompatibility Assay of Cu/DVDMS

The 293T cells were seeded into the 96-well plates with the density of 10^4^ cells/well and cultured for 12 h. DVDMS and Cu/DVDMS were added and co-cultured for 24 h. Then, the free DVDMS or Cu/DVDMS were wiped out by removing the culture medium and washed with PBS for three times. The cell viabilities of 293T cells were detected by using the 3-(4,5-dimethylthiazol-2-yl)-2,5-diphenyltetrazolium bromide (MTT, Sigma-Aldrich, St. Louis, MO, USA) assay. The definition of “Cell viability” was “(experimental group − blank group)/(control group − blank group) × 100% ”.

### 2.7. In Vitro Toxicity of Cu/DVDMS

The 4T1 cancer cells were seeded into the 96-well plates with the density of 10^4^ cells/well and cultured for 12 h. DVDMS and Cu/DVDMS were added and co-cultured for another 12 h. Then, the free DVDMS or Cu/DVDMS were wiped out by removing the culture medium and washed with PBS for three times. The cells were exposed to US (1 MHz, 1 W/cm^2^, 50% duty, 3 min) for 3 min. The cells were further cultured for 12 h and detected using the MTT assay. 

### 2.8. Animal Experiments

The animal studies were conducted according to the protocol approved by the Animal Care and Use Committee of Xiamen University, China. For in vivo study of the prepared Cu/DVDMS, female BALB/c mice (5 weeks old, ~20 g) were obtained and raised at the Animal Care and Use Committee (CC/ACUCC) of Xiamen University. The 4T1 cells (5 × 10^6^ cells/site) were implanted subcutaneously into the mice to construct the 4T1 tumor models. In vivo imaging was conducted when the tumors volume reached ~100 mm^3^ (about 7 days after implant). 

### 2.9. In Vivo Biodistribution of Cu/DVDMS

When the tumor volume of tumor-bearing mice reached ≈100 mm^3^, the mice were divided randomly into two groups: DVDMS or Cu/DVDMS. DVDMS or Cu/DVDMS (100 μg/mL 50 μL) were intratumorally injected (i.t.) into the relevant group. At 12 h post-injection, the mice were sacrificed and the tumor was harvested for TEM. The PET images were captured at different time points after injection.

### 2.10. In Vivo SDT/CDT Efficacy

The therapy experiments were carried out when the tumors volume reached ≈ 100 mm^3^. The 4T1 tumor mice model was divided into 5 groups at random, including saline control (i.t.), PBS+US (i.t.), DVDMS+US (i.t.), Cu/DVDMS (i.t.), and Cu/DVDMS+US (i.t.). The groups were injected with 50μL PBS, DVDMS (5 mg/kg), or Cu/DVDMS (DVDMS 5 mg/kg). After 12 h post-injection, for all mice in the US groups, the tumor areas were exposed to US (1 MHz, 1 W/cm^2^, 50% duty, 3 min). The tumor volume and body weights of mice were recorded every other day. Each tumor volume, based on the caliper measurements, was calculated using the following formula: tumor volume = length × width^2^ × 0.5. The mice were photographed at different times and euthanized at 14 days. These organs (heart, liver, spleen, lung, and kidney) were collected for the Haematoxylin and eosin (H&E) staining.

### 2.11. Histology Studies

The major organs (heart, liver, spleen, lung, and kidney) of all groups were removed for H&E staining and observed by digital microscopy. TdT-mediated-dUTP nick end labeling (TUNEL) of the tumor was captured.

### 2.12. Statistical Analysis

In this study, statistical analyses were conducted by the GraphPad Prism 8 software. The *p* values were calculated by analyzing data with the Student’s t-test or nonparametric test. Significance in statistical analysis was defined as *p* < 0.05.

## 3. Results and Discussion

This study explored the preparation process of Cu/DVDMS using Cu^2+^ to assist in the assembly of DVDMS. A novel sonosensitizer was synthesized via dropping a solution of CuCl_2_ into the DVDMS solution with the assistance of PVP which could control the growth of nanostructures [26,27], based on a coordination interaction between Cu^2+^ and the sulfonate radicals of DVDMS (Figure 1). It was found that simply dropping a solution of CuCl_2_ into the DVDMS solution triggered the assembly of significantly different nanostructures. Remarkably, increasing the [Cu^2+^]/[DVDMS] ratio resulted in a gradual change in the color of the DVDMS solution from yellow to clear, and precipitation gradually appears (Appendix A). The results showed that the copper ions could spontaneously assemble with DVDMS to form new materials. This spontaneous assembly may be due to the interactions between Cu^2+^ and carboxylate radicals from DVDMS as described before [17,28].

To further study the coordination and self-assembly between DVDMS and Cu^2+^, the content of copper in nanomaterials was detected through ICP-MS. The results demonstrated that the content of copper in 100 μg/mL nanomaterials is 57.2 μg/mL, and the molar ratio of copper to DVDMS in the final product was 4.1. The synthetic Cu/DVDMS were subjected to TEM scanning to observe the morphology of the nanoparticles. As shown in Figure 1A, the morphology of the nanoparticles was circular. The coordination between copper ions and carboxylate radicals from DVDMS may have formed the circular nanoparticles. As expected, the results of dynamic light scattering (DLS) revealed that the size of the nanoparticles was ≈180 nm, which was in accordance with the TEM measurement. The suitable size might endow the nanostructures with excellent cellular accumulation. Meanwhile, as shown in Figure 1B, it was revealed that the zeta potential of Cu/DVDMS owned a significant change. The fluorescence and UV-vis-NIR absorption of DVDMS and Cu/DVDMS were further researched. As shown in Figure 1C, after the coordination and self-assembly between DVDMS and Cu^2+^, the absorption peak of Cu/DVDMS displayed a redshift. The absorption peak at 371 nm decreased markedly, and new absorption peaks appeared at 399 nm, which were different from the values obtained from the porphyrin self-assembly based on the π–π interaction or hydrogen bonding [29]. These results also validated the successful preparation of a new nanomaterial. To further study the coordination and self-assembly of Cu/DVDMS, X-ray photoelectron spectroscopy (XPS) was performed. As shown in Figure 1D, the C1s spectrum of Cu/DVDMS possessed one peak at 284.8 eV, and the N1s spectrum of the Cu/DVDMS owned a peak at 398.21 eV. The O1s spectrum of Cu/DVDMS showed a peak at 531.92 eV, and the Cu 2p spectrum of the Cu/DVDMS had a single peak at 934.57 eV. It was revealed that DVDMS could self-assemble with Cu^2+^ to form nanoparticles.

The SDT properties of Cu/DVDMS were evaluated in solution by detecting the fluorescence intensity of the 2′,7′-dichlorodihydrofluorescein (DCFH) probe. DCFH was oxidized by ROS and rapidly emitted green fluorescence (Ex: 488 nm, Em: 525 nm). It was found that Cu^2+^ was able to enhance the SDT properties of DVDMS. As shown in Figure 1E, the fluorescence intensity produced by the nanoparticles was twice as high as that of the free DVDMS—which may be based on the enhanced intersystem crossing—and increased the triplet state of DVDMS via copper ions. To further study the CDT properties of Cu/DVDMS, the ·OH generation of the nanoparticles was validated with the addition of GSH and H_2_O_2_ using methylene blue (MB), which can be degraded by ·OH and usually selected as an indicator of ·OH generation [21]. As shown in Figure 1F, a significant decrease in absorption value was observed when MB was incubated with Cu/DVDMS (with GSH and H_2_O_2_). However, there were no apparent changes in absorption values after MB was incubated with DVDMS, GSH, and H_2_O_2_. It was suggested that a Cu-driven Fenton-like reaction can effectively produce ·OH and Cu/DVDMS possessed outstanding CDT properties.

4T1 cancer cells were utilized to investigate the cellular uptake of Cu/DVDMS. As shown in Figure 2C, the results of ICP-MS demonstrated that the content of copper in 4T1 tumor cells incubated with Cu/DVDMS increased to a maximum concentration at 3 h after incubation. This result implied that Cu/DVDMS exhibited excellent cellular uptake and that it could quickly pass through the cell membrane and enter the cytoplasm of cancer cells. Additionally, the distinguished intracellular accumulation effect of the Cu/DVDMS was suitable for further SDT study. In addition, the fluorescence of Cu/DVDMS was further studied by confocal laser scanning microscopy. As shown in Figure 2A and Appendix A, the fluorescence microscopy images of 4T1 cells after incubation with DVDMS showed strong red fluorescence, whereas the fluorescence of 4T1 cells was incubated with Cu/DVDMS was quenched, which confirmed Cu^2+^-induced fluorescence quenching of DVDMS [30]. These results imply that the nanomaterial could be efficiently delivered into the cytoplasm. To further study the intracellular localization of Cu/DVDMS in 4T1 cells, cells were observed by TEM. As shown in Figure 2B and Appendix A, the black nanoparticles were observed in the cell cytoplasm in cancer cells incubated with Cu/DVDMS, whereas no similar findings were observed in cancer cells incubated with DVDMS. These results showed that the nanoparticles could enter the cytoplasm of cancer cells easily compared with free DVDMS.

The intracellular ROS production capability of Cu/DVDMS was further investigated by DCFH-DA probe, which was oxidized by ROS and rapidly emitted green fluorescence (Ex: 488 nm, Em: 525 nm). The DCFH-DA probe was utilized and the fluorescence intensity was detected after the material was introduced into tumor cells. As shown in Figure 3A,B, the group treated with Cu/DVDMS produced more ROS than the groups treated with PBS or DVDMS, indicating the progression of a Fenton-like reaction. There was no apparent green fluorescence in the fluorescence microscopy images of the Cu/DVDMS without the US group, but flow cytometry analysis of cells stained with DCFH-DA confirmed elevated ROS production. Under US irradiation (1 W/cm^2^), the group treated with Cu/DVDMS+US exhibited the highest ROS production, because of the combination of SDT and CDT; this result was consistent with those of the ROS study of the nanostructures. It was demonstrated that the nanoparticles exhibited excellent cellular uptake and high ROS production from the enhanced SDT and Fenton-like reaction.

The biosafety of Cu/DVDMS was evaluated on 293T cells by using the typical 3-(4,5-dimethyl thiazol-2-yl)-2,5-diphenyltetrazolium bromide (MTT) protocol. The cell viability of 293T cells was detected after 24 h of incubation with Cu/DVDMS or DVDMS. As shown in Figure 3C, the prepared nanostructures showed no obvious cytotoxicity—even at a concentration of 40 µg/mL—compared with free DVDMS, demonstrating their excellent biocompatibility. Further in vitro SDT/CDT effects of Cu/DVDMS were evaluated on 4T1 cancer cells using the standard MTT assay. As shown in Figure 3D, cell viability decreased when the cells were treated with Cu/DVDMS, DVDMS+US, and Cu/DVDMS+US at different concentrations. The results showed that Cu/DVDMS possessed excellent CDT effects after the nanoparticles entered the cell, and the group treated with Cu/DVDMS+US possessed the best treatment effect, showing that Cu/DVDMS+US had an excellent capability for producing ROS damage in cancer cells.

To further evaluate the bio-distribution of Cu/DVDMS, an in vivo imaging study was conducted using a 4T1 tumor model. PET was performed at different time points after injection. As shown in Figure 4F and Appendix A, the signals of the Cu/DVDMS were mostly in the tumor section before 12 h and exhibited an obvious dispersion after 12 h [31]. TEM was also employed to study the bio-distribution. As shown in Figure 4B and Appendix A, the black nanoparticles, whose diameters were approximately 180 nm, were observed in the cytoplasm in cancer cells incubated with Cu/DVDMS.

We continued to investigate in vivo treatment effects by injecting Cu/DVDMS or DVDMS into different groups. The studies of different treatments were performed using 4T1 tumor-bearing mice, and in vivo tumor inhibition effect was evaluated by dynamically monitoring the changes in tumor volume. As shown in Figure 4A, in vivo therapeutic study was performed when the tumors volume reached ≈100 mm^3^ (approximately 7 days after inoculation). Cu/DVDMS or DVDMS was injected into the mice, and some groups were treated with ultrasound after 12 h post-injection at 1, 3, and 7 days, respectively. As shown in Figure 4C,E, the mice treated with PBS+US exhibited rapid tumor growth, which aligned with the results of the control group, indicating the insignificant antitumor effect of PBS+US groups. It was implied that ultrasound alone did not produce the inhibitory effects. The group treated with Cu/DVDMS and DVDMS+US could partially inhibit tumor growth within 14 days. The tumor volume of Cu/DVDMS+US group had a significant difference compared with Cu/DVDMS and DVDMS+US group, indicating the group treated with Cu/DVDMS+US possessed the optimal therapeutic effect relative to other treatment groups. As a whole, it was indicated that Cu/DVDMS can be applied as a nanomaterial for effective inhibition of tumor growth when exposed to US, which was related to excellent tumor accumulation, Fenton-like reaction, and greater ROS production of Cu/DVDMS.

As shown in Figure 4D, no significant body weight changes were found in any group during the 14 days, suggesting the low toxicity and safety of these therapies. Furthermore, hematoxylin and eosin (H&E) staining of major organs (heart, liver, spleen, lung, and kidney) were performed, revealing no obvious pathological changes in the tissues of all groups (Appendix A). These results indicated that Cu/DVDMS possessed high therapeutic biosafety.

Furthermore, H&E staining and TdT-mediated-dUTP nick end labeling (TUNEL) of tumors were performed to further study the therapeutic effect of the nanoparticles. As shown in Figure 4G, tumor tissues in the mice treated with Cu/DVDMS+US suffered more severe damage than these mice treated with PBS, PBS+US, DVDMS+US, or free Cu/DVDMS. In summary, these results clearly indicated that multifunctional Cu/DVDMS considerably improved tumor therapeutic efficacy.

## 4. Conclusions

In conclusion, we here reported the design and synthesis of a novel self-assembling Cu-porphyrin nanosonosensitizer based on DVDMS, which was prepared using a straightforward self-assembly procedure and able to enhance SDT. The copper ions and porphyrin inside the nanoparticles endowed this nanosonosensitizer with excellent PET imaging and SDT/CDT properties. The results showed that the nanoparticles could possess a synergetic combination of SDT and CDT. It was found that the nanoparticles could be efficiently delivered into the cytoplasm. In the tumor microenvironment, Cu/DVDMS was reduced by GSH and led to the secretion of Cu^+^ after the nanoparticles entered the cytoplasm of cancer cells. The cascades of bioreactions induced by copper not only depleted intracellular GSH, but also mediated a cytotoxic Fenton-like reaction to destroy tumor cells; furthermore, this nanostructure could enhance SDT. The high ROS production efficiency, high biosafety, and excellent therapeutic effect of Cu/DVDMS were demonstrated by the systematic in vivo and in vitro studies. Compared with other similar systems such as the PtCu_3_ nanocages from Zhong X. et al., we conducted elementary self-assembly based on DVDMS [8]. Because DVDMS is an active ingredient extracted from Photofrin and has been approved as a clinical drug by FDA, the self-assembly nanoparticles based on DVDMS had the potential for clinical transformation. We believe that this report can represent a simple approach to combining SDT and CDT.

## Data Availability

Not applicable.

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
