# Peer review of "Application of Self-Assembly Nanoparticles Based on DVDMS for Fenton-Like Ion Delivery and Enhanced Sonodynamic Therapy"

_biosensors, 2022, doi:10.3390/bios12040255_

Round 1

Reviewer 1 Report

The author reported manuscript titled " Application of self-assembly nanoparticles based on DVDMS for Fenton-like ion delivery and enhanced sonodynamic therapy ". This work is definitely of interest and novelty in self-assembly nanoparticles in SDT. The results are well presented and described. However, there are some concerns that should be addressed before its publication. Thus, I support the acceptance of this work after the authors address the following minor issues.

  1. In this manuscript, the combination of sonodynamic therapy and chemodynamic therapy were applied in cancer therapy. However, the authors didn’t review the previous studies of SDT and CDT combined treatment. I suggest the authors should add this part in “Introduction”.
  2. In this manuscript, we can see form Figure 2, the ICP results showed that the content of copper in 4T1 tumor cells incubated with Cu/DVDMS increased to a maximum concentration at 3 h after incubation and the fluorescence was quenched. However, there are weak fluorescence at 12 h. I suggest the authors should describe and discuss it more precisely in “Results”.
  3. The English writing of this manuscript need to be improved when revised.

Reviewer 2 Report

The authors presented a new type of nanoparticles Cu/DVDMS that can enhance both SDT/CDT functions. They have done extensive in vitro and in vivo experiments to demonstrate biosafety and tumour cytotoxicity. However, there are some major limitations of the current manuscript.

Major points:

  1. Intracellular dynamic of Cu/DVDMS. The authors mentioned Cu induces fluorescence quenching of DVDMS and this can recover over time but did not show any data. The inclusion of this information will enhance the manuscript and make the current statements stronger.
  2. There is no CDT only control (e.g., use CuCl2 solution with equivalent Cu concentration) in the experimental design. The inclusion of this control will be able to help the reader to further understand this new type of nanoparticles. For example, did the formation of NP affect the CDT capability?
  3. In the Discussion, the authors should discuss how their system can be transferred to clinical practice. Currently, the authors introduced the nanoparticle through intratumoral injection. However, this is not commonly done clinically. Also, the authors should discuss the pros and cons of the current Cu/DVDMS system over other similar systems (e.g., the PtCu3 nanocages from Zhong et al 2020 DOI: 10.1002/adfm.201907954).

There are also some minor points that required a further edition of the manuscript to improve the overall readability.

Minor points:

In the abstract, the authors mentioned ‘’DVDMS is a sonosensitizer approved by the FDA, but it has some deficiencies.’’ It will be great if the authors can specify some of the deficiencies that they are trying to address in this manuscript to give the reader a clear idea without having to dig into the content. The same happens at Line 43-44, ‘however, such photosensitizers have great limitations’ and the limitations were not mentioned in the next sentences.   

Also, in the abstract, the full name of DVDMS, sinoporphyrin sodium, should show up when it was first mentioned. The authors should also consider re-write their abstract as more sentences were used to introduce the background rather than describe the methods and results of their research.

The authors should make sure all the non-common abbreviations have been spelt out the first time they use it to avoid confusion. For example, “NPs” “US” in the Materials and Methods. Please add (i.t.) after ‘intratumorally injected’ in line 149. Also, they should provide the catalogue number and name of the suppliers for the chemicals and equipment in the Materials and Methods.

In Line 112-113, the authors mentioned the culture dish was washed three times with PBS and then stained for DAPI for confocal microscopy. Did the authors fix the cells in between? DAPI did not generally stain live cells as it is not cell membrane permeable unless on high concentration. If using high concentration, the authors should provide the concentration they used for the staining.

Please use the lower-case letter ‘k’ in the ‘kg’. ‘Kg’ is not the correct SI unit.

Line 161-162, ‘These organs (heart, liver, spleen, lung, and kidney) were collected for the Haematoxylin and eosin (H&E) staining.’ is repetitive to Line 165-167. Line 165, ‘Furthermore, the tissues of all groups were tested.’ is not necessary.

In the Result section, please point out the Figure panel when you indicate it in the sentences. For example, ‘Meanwhile, it was revealed that the zeta potential of Cu/DVDMS had a significant change’ should be followed by (Figure 1B).

In Figure 1, does NP = Cu/DVDMS? Can the authors mention how many replicates were done for each group on panels B and E? Also, is there a statistical significance between the DVDMS and Cu/DVDMS on those graphs?

In Figure 1D, what does each line of different colour means in those graphs? Also, the label ‘C1s’, ‘O1s’, etc are too small and burred to see.

Figure 2C, please indicate how many replicates was this and any statistical analysis has been done?

Line 240-241, ‘The fluorescence of 4T1 cells incubated with Cu/DVDMS recover slightly after 12 h.’ was mentioned but no image or data was shown, which made the latter statement about Cu/DVDMS separation in cytoplasm weak.

Line 247-248, ‘These results showed that the nanoparticles entered the cytoplasm of cancer cells easily compared with free DVDMS.’ This statement is very weak and not a logical comparison because TEM is not the right technique to detect free DVDMS. If you make this comparison, the fluorescent images in Figure 2A actually suggest otherwise since free DVDMS were more visible in the cell cytoplasm.    

Line 257-261, ‘the group treated with Cu/DVDMS produced more ROS than the groups treated with PBS or DVDMS, indicating the progression of a Fenton-like reaction. There was no apparent green fluorescence in the fluorescence microscopy images of the group treated with Cu/DVDMS, but flow cytometry analysis of cells stained with DCFH-DA confirmed ele-vated ROS production’ It is not clear which group the sentence was described. (-) US or (+) US?

The scale bar in Figure 3A is not visible. In Figure 3C and 3D, please indicate how many biological replicates were done for each group, how the 100% cell viability is defined, and what kind of statistical analysis was used. The statistical significance sign ** may need to be re-plotted as they are either misaligned or not clear positioned.

Line 280, the authors mentioned ‘The results showed that Cu/DVDMS had good CDT effects after the 280 nanoparticles entered the cell,’ This conclusion was more from the TEM result. I do not think the MTT assay result reflected that. Also, your TEM result was from 12 hours, not for 24 hours like the MTT. Any idea if you still see NPs in cells at a similar incubation time from MTT?

Please use arrows or red circles to point out where the nanoparticles are in Figures S2 and S3.

Please indicate what the error bar in the bar graphs is. Standard error or standard error of the mean?

The legend of y-axis of Figure 4D is wrong. Please check.

Any statistics has been conducted on the data of Figure 4C? If so, please show the result to support the statement from Line 304-313. Please also indicate how many mice were used for each group in this experiment.

Figure 4F, what were the non-tumour organs bright up at the 12 hr time point?

Figure 4G, were the tumours collected 12 hours post-treatment or 14 days post-treatment?

Reviewer 3 Report

The manuscript titled “Application of self-assembly nanoparticles based on DVDMS for Fenton-like ion delivery and enhanced sonodynamic therapy” by Jinqiang Liu et al focuses on understanding the application of sinoporphyrin sodium (DVDMS) and copper ion to produce a synergetic effect of sonodynamic therapy (SDT) and chemo dynamic therapy (CDT) for effective cancer treatment. In particular, this research aims to 1) produce self-assembled nanoparticles (NPs) by coordinating DVDMS and copper (Cu/DVDMS NPs) with the assistance of PVP, 2) investigate the safety profile and effectiveness of Cu/DVDMS NPs in vitro and in vivo. Several points need to be addressed before publication.

Major

Figure 1: For the characterization of the fabricated nanoparticles with/without cooper.  

  1. The author described that the sonosensitizer was fabricated with the help of PVP. The role of PVP in the process should be explained.
  2. The author claimed that in Figure 1A and 1B, the morphology of nanoparticles was oval. However, the particles shown in figure 1A look circular rather than oval, and figure 1B does also does not support what the authors claimed.
  3. The figures are too small. It should be enlarged to enhance the readability.

Figure 2:

  1.  To investigate the cellular uptake of NPs in the cancer cells, the authors performed LSCM. However, the images showed only a few 4T1 cells incubating with DVDMS and Cu/DVDMS after 12h and claimed the nanomaterial could be efficiently delivered into the cytoplasm. This is insufficient evidence due to the lack of quantitative data. The authors need to add a quantitative analysis of the results. Moreover, it is recommended to provide an image of a group of cells.
  2. The authors claimed that the fluorescence signals of 4T1 cells incubated with Cu/DVDMS recovered slightly after 12 h (line 240-241) but there is evidence to support it.

Figure 3:

  1. The scale bar is not noticeable.
  2. The authors used “relative viability”. It needs more explanation of how it was defined. Instead of using relative viability, statistical analysis using original viability values is recommended.
  3. The authors conclude their nanoparticle systems work cancer cell-specifically by testing only one cancer cell line and one normal cell line. More test results using other cell lines are required to reach the conclusion.

Figure 4:

  1. How many mice were used in the test? Sample numbers and statistical results are completely missing.
  2. Quantitative analysis of PET and TUNEL assay results may be helpful to strengthen what the authors claimed.
  3. Description in the main body (body weight) and legends in Figure 4D (relative tumor volume) are not corresponding. Please check this.
  4. In figure 4C, NP and DVDMS with US are relatively effective to inhibit the growth of tumors compared to control groups. However, as time goes by, the tumor size got larger and a reduction in tumor size was not found in all conditions. This reviewer wonders how the author can claim their nanoparticle and ultrasound-based technique can be utilized for cancer treatment. Further discussion is required.

Additionally, 

  1. Abbreviation should come after the full name.
  2. The materials and methods section should include more specific details including vendor, catalog number of each reagent, equipment's model, and statistics.
  3. A language check is recommended.

Round 2

Reviewer 2 Report

The authors have taken the time to address all my comments but failed to reflect some of them in the manuscript. For example, the figure legends still miss the n number and by only reading the manuscript, it is still not clear the error bar means SD or SEM. There is no discussion of the benefit of the models over other similar systems. The authors should embed their answers in the manuscript so the readers can have a full picture of their research. Also, I suggest changing the 'Results' section to 'Results and Discussion' and changing the 'Discussion' into 'Conclusion' to better reflect the writing content.  

Reviewer 3 Report

  1. The authors cited a few references about the function of PVP. However, the role of PVP in these references ( Prog.,2015, Vol. 31, No. 5 and Adv. Mater. 2017, 29, 1605928) is not clearly described. In addition, the other references (ACS Appl. Mater. Interfaces and Nanoscale) used the PVP for the synthesis of Au NPs. Thus, this reviewer is curious whether the PVP works in the same way for the Cu-based NPs synthesis or not. The author is recommended to explain the function of PVP in detail and cite more relevant references.

  1. Figure S5. Time-lapse images of a group of cells are required to prove the recovery.

  1. Figure 3. Regarding cell viability. The definition of cell viability should be noted in either in Method section or the main body.

  1. Figure 4, the number of mice number and their statistical results are still missing.

  1. The PET and TUNEL assay results in the authors’ response document should be included in the manuscript.

Minor

  1. The authors should be consistent with the usage of abbreviations in the manuscript. For example, “CLSM” and “LSCM”.
  2. Statistical methods and software that the authors used should be written in the “Materials and Methods” section.
  3. Again, a language re-check is recommended.
